

# Molecular profiling and sex determination of *Cannabis sativa* germplasm collection: Exploring microsatellite markers and high-resolution melting (HRM) analysis

Anastasia Boutsika[1,*], Eleftheria Deligiannidou[1,*], Theodoros Moysiadis[1,2], Nikolaos Tourvas[1], Panagiotis Karnoutsos[1,3], Marios Karagiovanidis[1,3], Dimitrios Magalios[4], Christos Nanos[4], Vangelis Mitsis[5], Eleni Tsaliki[1], Eirini Sarrou[1], Apostolos Kalivas[1] and Ioannis Ganopoulos[1]

[1] ELGO DIMITRA, Institute of Plant Breeding and Genetic Resources, Thessaloniki, Greece
[2] Department of Computer Science, School of Sciences and Engineering, University of Nicosia, Nicosia, Cyprus
[3] IA Agro S.M.P.C., Thessaloniki, Greece
[4] IG Agrotech P.C., AlmaCann, Athens, Greece
[5] Ekati Alchemy Lab SL, Moià, Spain
[*] These authors contributed equally to this work.

Corresponding author
Ioannis Ganopoulos,
giannis.ganopoulos@gmail.com

## ABSTRACT

*Cannabis sativa* L., a versatile plant grown for its seeds, fiber, and cannabinoids, has recently received significant scientific interest due to its wide range of industrial and pharmaceutical uses and economic prospects. The objective of this study is to assess the genetic variation of cannabis by examining its morphological and molecular characteristics in 83 different genotypes preserved at the Institute of Plant Breeding and Genetic Resources in Thessaloniki. Utilizing 10 microsatellite markers, important genetic variation was observed among the samples. Population structure analysis using STRUCTURE software indicated four distinct genetic subpopulations, which were further supported by the principal coordinate analysis (PCoA). The validity of these clusters was further confirmed through unweighted pair group method with arithmetic mean (UPGMA) phylogenetic analysis. The analysis of molecular variance (AMOVA) indicated that most of the genetic variation is present within individuals rather than among them or among populations (9% among populations, 53% within individuals). The expected heterozygosity (He or uHe) turned out to be 0.60–0.66, which supports moderate diversity. The fixation indices (Fst = 0.094, Fis = 0.417, Fit = 0.472) were statistically significant and suggested a moderate level of genetic differentiation between the two populations. Nei's genetic distance between hemp and marijuana groups was calculated as D = 0.288, indicating a moderate level of genome-wide divergence between these two major use types. Additionally, high-resolution melting (HRM) analysis utilizing sex-linked markers successfully identified 73 female and 10 male plants, underscoring the value of early sex identification in enhancing breeding strategies. These findings contribute valuable insights into the genetic diversity and sex determination mechanisms of *Cannabis sativa* and support future breeding programs which aim at developing cultivars with favorable traits.

## INTRODUCTION

*Cannabis sativa* is an annual medicinal plant of the Cannabaceae family of the order Urticales. Being among the oldest economic plants, this multipurpose crop has been used for fiber and food production and has many compounds that have been used therapeutically (*Toth et al., 2020*; *Ross, 2023*). *Cannabis sativa* is diploid (2n = 20) with a karyotype composed of nine autosomes and a pair of sex chromosomes (*Moliterni et al., 2004*; *Razumova et al., 2015*).

Delta-9 tetrahydrocannabinol (THC), the primary psychoactive compound in *C. sativa*, has been the most studied cannabinoid due to its well-known effects on cognition, appetite, and mood (*Fearby, Penman & Thanos, 2022*). Recently, interest has grown in non-intoxicating cannabinoids such as cannabidiol (CBD), which has been associated with antipsychotic, anxiolytic, and anti-craving effects (*Batalla et al., 2021*). In addition, other cannabinoids, including tetrahydrocannabinolic acid (THCA), cannabidiolic acid (CBDA), cannabigerolic acid (CBGA), cannabigerol (CBG), cannabinol (CBN), tetrahydrocannabivarinic acid (THCVA), and cannabidivarinic acid (CBDVA), are gaining attention for their potential therapeutic properties (*Walsh, McKinney & Holmes, 2021*; *Udoh et al., 2022*).

Internationally, the most significant classification from a legal and commercial standpoint revolves around the concentration of the primary psychotropic compound, THC (*Small, 2015*). Hemp is cultivated for industrial purposes (*e.g.*, fiber, seed, oil) or for cannabinoid extraction (primarily CBD), and is defined by a very low THC content, typically $\leq$ 0.3% THC by dry weight in the USA and Canada, or $\leq$ 0.2% THC in much of Europe (*Schluttenhofer & Yuan, 2017*; *Johnson, 2019*). Marijuana, on the other hand, is cultivated for its higher content in THC, which is > 0.3% (or > 0.2%) on a dry weight basis, and can range between 5%–30% in specialized cultivars intended for medicinal or recreational psychoactive use (*ElSohly et al., 2016*). However, *Russo (2019)* supported the classification of *C. sativa* based on its 'chemovar' (chemical variety) focusing on the dominant cannabinoids and often the terpene profile as more accurate and scientifically sound. As a result, the following chemotypes were adopted: Type I, which is THC-dominant and low in CBD ('Marijuana'), Type II, which exhibits balanced THC and CBD (mixed ratio, often sought for specific medicinal applications), Type III, which is CBD-dominant and low in THC ('Hemp'), and includes specific cultivars bred for high CBD and low THC for medicinal use without strong psychoactive effects), Type IV, which is CBG-dominant with low THC (emerging chemovars), and Type V, which contains no detectable cannabinoids (zero-cannabinoid, primarily experimental or theoretical at this stage for industrial fiber).

*Cannabis sativa* is primarily a dioecious species, meaning that male and female reproductive organs are found on separate plants, a trait observed in only about 5–6% of angiosperms (*Baek & Vergara, 2025*). Female plants are typically cultivated for their flowers, which are rich in glandular trichomes that synthesize and store phytocannabinoids
and other phytochemicals (*Happyana et al., 2013*). Male plants are usually culled from cannabinoid-focused cultivation to prevent pollination, which leads to seeded flowers and reduced quality (*Feder et al., 2021*; *Valizadehderakhshan et al., 2021*; *Todd, Song & Van Acker, 2022*; *Torres et al., 2022*). However, males are used in fiber production, grain and seed oil extraction, and breeding programs (*Flajšman, Slapnik & Murovec, 2021*; *Valizadehderakhshan et al., 2021*). According to *Lapierre, Monthony & Torkamaneh (2023)*, there are several competing theories on the taxonomy of *Cannabis* species. The idea of a highly varied monotypic species is strongly supported by new data from modern genomic sequencing and other molecular methods. Although some researchers consider the genus *Cannabis* to be polytypic (*Hillig & Mahlberg, 2004*; *Hillig, 2005*; *Clarke & Merlin, 2014*), Plants of the World Online (*Plants of the World Online, POWO)(2024*) recognizes only a single species, *C. sativa*. However, *Plants of the World Online (POWO)(2024)* lists up to 36 heterotypic synonyms, reflecting historical and regional taxonomic differences. Furthermore, *C. sativa* has a long history as a medicinal plant with high ethnopharmacological significance (*Bonini et al., 2018*).

The origin of *C. sativa* remains a subject of debate. Its evolutionary roots were once thought to lie in temperate regions of western or central Asia (*Small, 2017*). However, reports have also proposed a multiregional origin rather than a single geographic source (*Long et al., 2017*), and the northeastern Tibetan Plateau has been widely accepted among experts as a probable center of origin based on fossil pollen data (*McPartland, 2018*). Some evidence supports early domestication in East Asia, with modern cultivars descending from an ancestral gene pool that is now represented by landraces and feral plants in China, and additional plastome data points to Yunnan as a likely region of origin (*Ren et al., 2021*; *Osterberger et al., 2021*).

Identification of genetic variability among *C. sativa* genotypes is essential for crop improvement, and molecular markers play a key role in selecting and differentiating diverse lines (*Gao et al., 2014*). Numerous studies using various marker types have confirmed that *C. sativa* is highly genetically diverse (*Lynch et al., 2016*; *Punja, Rodriguez & Chen, 2017*; *Soler et al., 2017*; *Henry et al., 2020*; *Zhang et al., 2020*; *Pandey et al., 2023*). The release of the first draft genome and transcriptome (*van Bakel et al., 2011*) significantly advanced cannabis research and enabled a shift from traditional to molecular breeding approaches. Subsequent draft assemblies have further supported progress in medicinal cannabis research, including genomics, pan-genome analysis, genomic selection, and genome editing (*Braich et al., 2020*; *Grassa et al., 2021*; *Ryu et al., 2024*; *Lynch et al., 2025*). In parallel, marker-assisted selection has already been applied successfully using simple sequence repeat (SSR) markers, which do not require reference-quality genomes and remain a valuable tool for breeding applications (*Barcaccia et al., 2020*).

*Cannabis sativa* is considered a valuable model species for studying plant sex determination due to its distinct XY chromosome system and a long history of cytogenetic research (*Karlov et al., 2017*). Sex is a key trait in breeding programs (*Moliterni et al., 2004*), with its determination being complex, because it is influenced by both genetic and environmental factors, although genetic influences are generally predominant (*Schilling et al., 2021*). This presents a challenge in breeding, as sex can typically only be confirmed at

the onset of flowering when sexual dimorphism becomes visible (*Barcaccia et al., 2020*). To address this, early identification using reliable sex-linked, Y-specific DNA markers is essential (*Moliterni et al., 2004*; *Barcaccia et al., 2020*). High-resolution melting (HRM) analysis offers a rapid, cost-effective method for identifying sex-linked single nucleotide polymorphisms (SNPs). It is widely used in plants for the detection of nucleotide polymorphisms due to its high sensitivity, efficiency, and reduced time requirements compared to other post-PCR genotyping techniques (*Ganopoulos et al., 2014*; *Taheri et al., 2017*). In this study, HRM was used to distinguish SNP differences between male and female cannabis plants (*Gilchrist et al., 2021*).

Given the importance of genetic diversity in medicinal plants for trait discovery, and the influence of plant sex on secondary metabolite production in *C. sativa*, this study aims to characterize genetic variation among 83 accessions using 10 SSR markers, and to identify plant sex using HRM. These tools provide actionable data to improve germplasm management and inform future breeding strategies.

## MATERIALS & METHODS

### Plant material

Sampling was conducted from the germplasm of *C. sativa* lines preserved and/or developed for breeding purposes at the Institute of Plant Breeding and Genetic Resources (IPGRB, ELGO-DIMITRA) in Thessaloniki, Greece. The study included 83 *C. sativa* plants, representing both hemp and marijuana samples, diverse in cannabinoid qualitative/quantitative content, as presented in Table 1. The samples were classified into four chemotypes: Type I, Type II, Type III, and Type IV). Types III and IV refer to hemp, while Types I and II refer to marijuana.

### Morphological trait determination

Initial sexing was based on the presence of pistillate or staminate flowers, and these phenotypic observations were used as reference data to validate the HRM-based sex identification. No monoecious or intersex (hermaphrodite) plants were observed.

### Extraction and liquid chromatographic analysis

Extraction was performed by mixing 0.1 g of dried and milled *C. sativa* inflorescences with five mL methanol in an orbital shaker for 15 min. The mixture was then centrifuged for 5 min at 500 g at 4 °C, and the supernatant was transferred to a clean volumetric flask. The procedure was repeated twice, and all supernatants were combined, filtered through a 0.45 µm membrane filter into 2 mL glass vials, and injected directly for analysis.

The analysis was performed using a Shimadzu Nexera HPLC system (Kyoto, Japan), which consisted of two LC-30AD pumps, a DGU-20A5 degasser, a CTO-20AC column oven, a SIL-30AC auto injector, and an SPD-M40 diode array detector (DAD). Cannabinoid seperation was performed according to *Galettis et al. (2021)*, using a NexLeaf CBX for Potency column (2.7 µm, 4.6 ×150 mm) with a pre-column temperature set at 35 °C and a flow rate of 1.4 mL/min. The injection volume was 5 µL and chromatographic separation was achieved in 11 minutes using an isocratic method with 5 mM ammonium formate

**Table 1** *Cannabis sativa* sample coding, cannabinoid profile, origin classification, and gender.

| Number | Sample coding | Type | Cannabinoid profile | Pre-defined population | Gender (HRM) |
|---|---|---|---|---|---|
| 1 | A.18.1 | Hemp | CBD: 5.30% THC: 0.26% | Type III | Female |
| 2 | A17.1 | Hemp | CBD: 7.30% THC: 0.29% | Type III | Female |
| 3 | A20.8 | Hemp | CBD: 7.80% THC: 0.29% | Type III | Female |
| 4 | A29.5 | Hemp | CBD: 5.60% THC: 0.29% | Type III | Female |
| 5 | A35.1 | Hemp | – | Type III | Male |
| 6 | A36.1 | Hemp | CBD: 5.50% THC: 0.26% | Type III | Female |
| 7 | A7.1 | Hemp | CBD: 4.80% THC: 0.25% | Type III | Female |
| 8 | A7.3 | Hemp | – | Type III | Male |
| 9 | A7.6 | Hemp | CBD: 4.40% THC: 0.23% | Type III | Female |
| 10 | B51 | Hemp | CBG: 5.50% THC: 0.08% | Type IV | Female |
| 11 | B53 | Hemp | CBG: 6.00% THC: 0.14% | Type IV | Female |
| 12 | B53.N2.16 | Hemp | CBG: 3.10% THC: 0.10% | Type IV | Female |
| 13 | B53.N2.16 | Hemp | CBG: 3.25% THC: 0.09% | Type IV | Female |
| 14 | B53.N2.16 | Hemp | CBG: 3.20% THC: 0.09% | Type IV | Female |
| 15 | B53.N2.22 | Hemp | CBG: 2.10% THC: 0.07% | Type IV | Female |
| 16 | B53.N2.22 | Hemp | CBG: 2.00% THC: 0.06% | Type IV | Female |
| 17 | B53.N2.22 | Hemp | CBG: 2.30% THC: 0.07% | Type IV | Female |
| 18 | B53xB53N28 | Hemp | CBG: 4.80% THC: 0.13% | Type IV | Female |
| 19 | B53xB53N28 | Hemp | CBG: 4.60% THC: 0.12% | Type IV | Female |
| 20 | B53xB53N28 | Hemp | – | Type IV | Male |
| 21 | BIALOBRZESKIE | Hemp | CBD: 1.80% THC: 0.16% | Type III | Female |
| 22 | CARMAGNOLA | Hemp | CBD: 1.85% THC: 0.14% | Type III | Female |
| 23 | CB611 | Hemp | CBG: 5.00% THC: 0.07% | Type IV | Female |
| 24 | CG10C1 | Marijuana | CBD: 0.15% THC: 24.88% | Type I | Female |
| 25 | CG15C1 | Marijuana | CBD: 0.13% THC: 25.00% | Type I | Female |
| 26 | CG19C2 | Marijuana | CBD: 0.15% THC: 26.24% | Type I | Female |
| 27 | CG22C1 | Marijuana | CBD: 0.15% THC: 26.67% | Type I | Female |
| 28 | CG31C2 | Marijuana | CBD: 0.13% THC: 24.06% | Type I | Female |
| 29 | CG36c1 | Marijuana | CBD: 0.15% THC: 22.39% | Type I | Female |
| 30 | CG38C2 | Marijuana | – | Type I | Male |
| 31 | CG3C1 | Marijuana | CBD: 0.15% THC: 26.07% | Type I | Female |
| 32 | CG4C1 | Marijuana | CBD: 0.14% THC: 25.06% | Type I | Female |
| 33 | CS | Hemp | CBD: 1.11% THC: 0.15% | Type III | Female |
| 34 | D1.2 | Hemp | CBD: 4.80% THC: 0.29% | Type III | Female |
| 35 | D7.1 | Hemp | CBD: 4.20% THC: 0.25% | Type III | Female |
| 36 | D7.1.29 | Hemp | CBD: 4.40% THC: 0.26% | Type III | Female |
| 37 | D7.1.29 | Hemp | CBD: 5.00% THC: 0.27% | Type III | Female |
| 38 | D7.1.29 | Hemp | CBD: 5.00% THC: 0.27% | Type III | Female |
| 39 | D7.7 | Hemp | CBD: 5.20% THC:0.29% | Type III | Female |
| 40 | D7.7.36.1 | Hemp | – | Type III | Male |

**Table 1** (*continued*)

| Number | Sample coding | Type | Cannabinoid profile | Pre-defined population | Gender (HRM) |
|---|---|---|---|---|---|
| 41 | D7.7.36.2 | Hemp | CBD: 5.30% THC: 0.29% | Type III | Female |
| 42 | D7.7.36.3 | Hemp | CBD: 5.00% THC: 0.29% | Type III | Female |
| 43 | D7.7.49 | Hemp | – | Type III | Male |
| 44 | D7.7.49 | Hemp | CBD: 5.10% THC: 0.28% | Type III | Female |
| 45 | D7.7.49 | Hemp | CBD: 5.20% THC: 0.29% | Type III | Female |
| 46 | FEDORA 17 | Hemp | CBD: 1.65% THC: 0.12% | Type III | Female |
| 47 | FELINA32 | Hemp | CBD: 2.65% THC: 0.07% | Type III | Female |
| 48 | FERIMON | Hemp | CBD: 1.25% THC: 0.11% | Type III | Female |
| 49 | FINOLA | Hemp | CBD: 1.05% THC: 0.11% | Type III | Female |
| 50 | FMC01 | Hemp | CBD: 4.25% THC: 0.28% | Type III | Female |
| 51 | FMc02 | Hemp | CBD: 1.05% THC: 0.11% | Type III | Female |
| 52 | FUTURA75 | Hemp | CBD: 3.46% THC: 0.13% | Type III | Female |
| 53 | G0c04 | Hemp | CBD: 3.46% THC: 0.21% | Type III | Female |
| 54 | GOC03 | Hemp | CBD: 4.90% THC: 0.29% | Type III | Female |
| 55 | H14C2 | Marijuana | CBD: 7.87% THC: 14.70% | Type II | Female |
| 56 | H1C1 | Marijuana | CBD: 0.18% THC: 22.55% | Type I | Female |
| 57 | H27 C1 | Marijuana | CBD: 0.14% THC: 28.42% | Type I | Female |
| 58 | H29 C1 | Marijuana | CBD: 0.12% THC: 25.31% | Type I | Female |
| 59 | I7.1 | Hemp | CBD: 4.12% THC: 0.28% | Type III | Female |
| 60 | K1.2 | Hemp | CBD: 3.70% THC: 0.28% | Type III | Female |
| 61 | KACO1 | Hemp | CBD: 4.17% THC: 0.26% | Type III | Female |
| 62 | KC DORA | Hemp | CBD: 2.17% THC: 0.22% | Type III | Female |
| 63 | KOMPOLTI | Hemp | CBD: 3.04% THC: 0.16% | Type III | Female |
| 64 | NWo1 | Hemp | CBD: 5.15% THC: 0.28% | Type III | Female |
| 65 | OTc10 | Marijuana | CBD: 0.11% THC: 23.49% | Type I | Female |
| 66 | OTc11 | Marijuana | CBD: 0.12% THC: 26.02% | Type I | Female |
| 67 | OTc15 | Marijuana | – | Type I | Male |
| 68 | OTc16 | Marijuana | CBD: 0.13% THC: 24.83% | Type I | Female |
| 69 | OTc17 | Marijuana | CBD: 0.13% THC: 24.17% | Type I | Female |
| 70 | OTc20 | Marijuana | CBD: 0.11% THC: 16.99% | Type I | Female |
| 71 | OTc23 | Marijuana | CBD: 0.12% THC: 20.90% | Type I | Female |
| 72 | OTc6 | Marijuana | CBD: 0.12% THC: 23.85% | Type I | Female |
| 73 | RH10C1 | Marijuana | CBD: 9.97% THC: 9.75% | Type II | Female |
| 74 | RH11C2 | Marijuana | CBD: 9.67% THC: 9.63% | Type II | Female |
| 75 | RH13C1 | Marijuana | CBD: 9.22% THC: 10.10% | Type II | Female |
| 76 | RH20C2 | Marijuana | CBD: 6.40% THC: 22.26% | Type I | Female |
| 77 | SANTHICA 27 | Hemp | CBG: 1.81% THC: 0.10% | Type IV | Female |
| 78 | SANTHICA70 | Hemp | – | Type IV | Male |
| 79 | STc04 | Hemp | – | Type III | Male |
| 80 | STc05 | Hemp | – | Type III | Male |
**Table 1** (*continued*)

| Number | Sample coding | Type | Cannabinoid profile | Pre-defined population | Gender (HRM) |
|--------|---------------|------|---------------------|------------------------|--------------|
| **81** | TIBORSZALLASI | Hemp | CBD: 1.46% THC: 0.11% | Type III | Female |
| **82** | TYGRA | Hemp | CBD: 1.31% THC: 0.11% | Type III | Female |
| **83** | USO 31 | Hemp | CBD: 1.10% THC: 0.11% | Type III | Female |

**Notes.**

The 'Sample coding' column contains unique identifiers assigned to each sample, which often include a combination of letters and numbers to denote specific characteristics or origins within the breeding program. The samples were classified regarding their 'Type' and specifically whether they are hemp or marijuana. The 'Cannabinoid Profile' column refers to the CBD and THC percentage of each sample. The 'Pre-defined Population' column refers to the Type of the sample in which case can be classified as one of the four types, which are specifically: Type I (THC-dominant and low CBD, marijuana), Type II (balanced THC and CBD mixed ratio, often sought for specific medicinal applications), Type III (CBD-dominant, low THC, hemp), Type IV (CBG-dominant, low THC, emerging chemovars). The 'Gender (HRM)' column reveals the sex that was identified by the HRM analysis and was either female or male.

and 0.1% formic acid (25%) and acetonitrile containing 0.1% formic acid (75%). For quantification of the major peaks, external standards of THC, THCA, CBD, CBDA, CBG, CBC, and CBN were used to develop calibration curves with concentrations ranging from 0.5 to 150 μg/mL. Detection was performed at 228 nm, and data acquisition and processing were carried out using Lab Solutions LC-MS software.

Although acidic precursors such as cannabigerolic acid (CBGA) and cannabichromenic acid (CBCA) can be detected with LC-MS, these compounds were excluded from the present analysis due to limitations related to sample stability and instrument optimization protocols. We acknowledge that CBGA, in particular, is essential for the accurate classification of Type IV chemovars, and future profiling efforts should include these precursors to better support chemotype differentiation.

## DNA extraction

Young leaf tissue was collected from each plant, immediately frozen in liquid nitrogen, and stored at −80 °C prior to DNA extraction. Genomic DNA was extracted using the CANVAX DNA Extraction Kit, following the manufacturer's protocol. DNA quality and quantity were assessed using a NanoDrop One/OneC spectrophotometer (Thermo Fisher Scientific) and confirmed by agarose gel electrophoresis.

## Microsatellite genotyping

Two multiplex PCR assays were designed with a total of 10 microsatellite markers (*Al-Ghanim, 2003*; *De Oliveira Pereira Ribeiro et al., 2020*; *Ioannidis et al., 2022*). The fluorophores used for the SSR markers were FAM, ROX, TAMRA, and HEX. The details of SSR markers, including their names, dye labels, amplicon size ranges (min-max), and sequences of forward (F) and reverse (R) primers are shown in Table 2. PCR reactions were performed in a total volume of 10 μL, consisting of 5 μL of KAPA2G Fast Multiplex (Kapa Biosystems), 1 μL of template DNA (50–100 ng/μL), 0.2 μL of forward and 0.2 μL of reverse primer mix (10 μM), and 2.4 μL of nuclease-free water. The thermal cycling conditions for PCR amplification for Multiplex 1 were: Initial denaturation at 95 °C for 5 min, denaturation at 95 °C for 30 s, annealing at specific temperatures for 30 s, and extension at 72 °C for 30 s, repeated for 35 cycles, final extension at 72 °C for 10 min, while for Multiplex 2: Initial denaturation at 94 °C for 5 min, denaturation at 94 °C for 30 s,

annealing at specific temperatures for 45 s, and extension at 72 °C for 1 min, repeated for 30 cycles, final extension at 72 °C for 7 min. Fragment analysis was conducted in an ABI 3730xl (Applied Biosystems, Foster City, CA, United States) with GeneScan 500 LIZ size standard and results were obtained and analyzed with GeneMapper v4 (Table S1).

## High-resolution melting (HRM) assay

Initial phenotypic sex determination was performed at the onset of flowering, based on the direct observation of reproductive organs. Plants were classified as male or female by the presence of staminate or pistillate flowers, respectively. These morphological assessments were used as reference standards to validate the results obtained by High-Resolution Melting (HRM) analysis.

To determine the sex of each sample, HRM analysis was performed by differentiating between the SNPs in male and female plants. After identifying two sex-specific SNPs near each other, *Gilchrist et al. (2021)* developed the primer pair EG160f and EG162r. These two sex-specific SNPs were homozygous in all females (A/A and G/G) and heterozygous in all males (A/G and G/C). HRM reactions were performed in a total volume of 25 μL. Specifically, 1 μL EG160f (10 μM), 1 μL EG162r (10 μM), 0.4 μL SYTO™ Green Fluorescent Nucleic Acid Stains (Thermo Fisher Scientific), 2.5 μL Buffer (Kapa Biosystems), 1 μL dNTPs (Kapa Biosystems), 0.2 μL Taq Polymerase (Kapa Biosystems) and 17.9 μL nuclease-free water were used for each sample. The thermocycling conditions were: Initial denaturation at 95 °C for 3 min, denaturation at 95 °C for 30 s, annealing at 55 °C for 30 s, and extension at 72 °C for 1 min, repeated for 40 cycles, followed by final extension at 72 °C for 10 min. Finally, the HRM analysis was carried out by the QuantStudio® 5 System (Thermo Fisher Scientific) and the HRM conditions were: 95 °C for 10 s, 60 °C for 1 min, 95 °C for 15 s and 60 °C for 15 s.

## Data analysis

The allelic data obtained from fragment analysis were statistically analyzed with GenAlEx (*Peakall & Smouse, 2012*), STRUCTURE software (*Pritchard, Stephens & Donnelly, 2000*), and RStudio version 4.3.1 (*R Core Team, 2021*). In RStudio, the following packages were used: ape (*Paradis & Schliep, 2019*), phangorn (*Schliep, 2011*), readr (*Wickham, Hester & Bryan, 2018*), ggplot2 (*Wickham, 2016*), ggrepel (*Slowikowski, 2021*), cluster (*Maechler et al., 2023*), factoextra (*Kassambara & Mundt, 2020*), and NbClust (*Charrad et al., 2014*).

The 'admixture' and 'independent allele frequencies' models, were used to run STRUCTURE 2.3.4 (*Pritchard, Stephens & Donnelly, 2000*). A burn-in of 200,000 iterations and 500,000 MCMC repetitions for each run were executed, with 20 replicates from $K = 1$ up to $K = 8$. The CLUMPAK main pipeline (*Kopelman et al., 2015*) was used to merge replicate runs, and the optimal K value was inferred using Evanno's method (*Evanno, Regnaut & Goudet, 2005*), which was run in the pophelper 2.3.0 R package (*Francis, 2017*). The software Structure threader (*Pina-Martins et al., 2017*) was used to parallelize computations. Principal coordinate analysis (PCoA) results were visualized using ggplot2, and the optimal number of clusters was determined using the Elbow method, Silhouette method, and gap statistic. The genetic distance matrix was calculated using

Boutsika et al. (2025), *PeerJ*, DOI 10.7717/peerj.19770
**Table 2   Multiplex PCR analysis, primers and probe characteristics.**   We performed two multiplex PCR reactions: Multiplex 1 and Multiplex 2, each having a combination of different markers stated below. From the bibliography we also obtained information regarding the dye, motif, type of repeat, min and max size of the amplicons, as well as the sequence of the forward (F) and the reverse primer (R).

| PCR | Name | Bibliography | Dye | Motif | Type of repeat | Min | Max | F | R |
|---|---|---|---|---|---|---|---|---|---|
| Multiplex 1 | C11-CANN1 | *Al-Ghanim (2003)*, *De Oliveira Pereira Ribeiro et al. (2020)* and *Ioannidis et al. (2022)* | FAM | (TGA)x(TGG)y | Compound/ Indel | 138 | 180 | GTGGTGGTGATGATGATAATGG | TGAATTGGTTACGATGGCG |
| | B02-CANN2 | *Al-Ghanim (2003)* | TAMRA | (AAG) | Simple | 163 | 172 | CAACCAAATGAGAATGCAACC | TGTTTTCTTCACTGCACCC |
| | ANUCS301 | *De Oliveira Pereira Ribeiro et al. (2020)* | FAM | (TTA) | Simple | 209 | 261 | ATATGGTTGAAATCCATTGC | TAACAAAGTTTCGTGAGGGT |
| | B05-CANN1 | *De Oliveira Pereira Ribeiro et al. (2020)* and *Ioannidis et al. (2022)* | HEX | (TTG) | Simple | 217 | 243 | TTGATGGTGGTGAAACGGC | CCCCAATCTCAATCTCAACCC |
| | ANUCS202 | *Ioannidis et al. (2022)* | HEX | (GA) | Simple | 145 | 185 | AGGACCAATTTTGAATATGC | AGAGAGGGAAGGGCTAACTA |
| | H09-CANN2 | *De Oliveira Pereira Ribeiro et al. (2020)* and *Ioannidis et al. (2022)* | TAMRA | (GA) | Simple | 204 | 224 | CGTACAGTGATCGTAGTTGAG | ACACATACAGAGAGAGCCC |
| Multiplex 2 | ANUCS305 | *De Oliveira Pereira Ribeiro et al. (2020)* and *Ioannidis et al. (2022)* | ROX | (TGG) | Simple | 125 | 170 | AAAGTTGGTCTGAGAAGCAAT | CCTAGGAACTTTCGACAACA |
| | ANUCS304 | *Ioannidis et al. (2022)* | FAM | (TCT)xTCA(TCT)y | Complex | 141 | 222 | TCTTCACTCACCTCCTCTCT | TCTTTAAGCGGGACTCGT |
| | ANUCS201 | *Ioannidis et al. (2022)* | HEX | (GA) | Simple | 155 | 227 | GGTTCAATGGAGATTCTCGT | CCACTAAACCAAAAGTACTCTTC |
| | B01-CANN1 | *De Oliveira Pereira Ribeiro et al. (2020)* and *Ioannidis et al. (2022)* | FAM | (GAA)x(A)(GAA)y | Complex | 323 | 339 | TGGAGTCAAATGAAAGGGAAC | CCATAGCATTATCCCACTCAAG |

the ape package, and the most distant sample was identified. A phylogenetic tree was constructed using the unweighted pair group method with arithmetic mean (UPGMA), implemented in the phangorn package, with the tree rooted using an outgroup and colored based on clustering. For determining the optimal number of clusters (K), the web-based software StructureSelector was used (*Li & Liu, 2017*). This tool accelerated the selection and visualization of the optimal number of clusters using multiple methods (*Pritchard, Stephens & Donnelly, 2000*; *Evanno, Regnaut & Goudet, 2005*; *Raj, Stephens & Pritchard, 2014*; *Kopelman et al., 2015*; *Puechmaille, 2016*).

## RESULTS

### Genetic diversity and population structure analysis

In contrast to the binary hemp *vs.* marijuana divide, STRUCTURE analysis (Fig. 1) showed four genetically distinct clusters with limited admixture ($K = 4$), which were also supported by $\Delta$K as the optimal number of clusters (Fig. 2), indicating a more complex population structure. The SSR data capture neutral evolutionary divergence rather than chemotype-driven selection, as evidenced by the notable fact that these clusters do not strictly correspond to chemovar types (Types I–IV). Agronomic selection for industrial traits most likely shaped the lineage corresponding to European fiber-type hemp, represented by the first group (Cluster A, black color) in the plot when viewed from left to right. A varied collection of accessions with different chemotype backgrounds can be found in the second cluster (Cluster B, orange), which seems to represent intermediate or admixed hybrids. The third group (Cluster C, blue) is primarily composed of marijuana accessions, including balanced Type II cultivars, which were probably chosen for their high cannabinoid content. Finally, the fourth group on the right (Cluster D, green) includes several narrow-spectrum, high-THC cultivars that demonstrate rigorous selection for psychoactive potency. The discovery of several subpopulations, especially within hemp, supports a multifaceted understanding of *C. sativa* genetic diversity by highlighting the existence of different breeding lineages or geographic origins.

### Principal coordinate analysis (PCoA) and determination of optimal number of clusters

Four major genetic clusters were further supported by PCoA (Fig. 3). 34.5% of the total genetic variation was explained by the first two axes (20.1% by PCoA1 and 14.4% by PCoA2). The analysis of the PCoA results and the association with Clusters A, B, C, and D, as shown in the STRUCTURE analysis, will be further explained later.

### UPGMA phylogenetic analysis

The UPGMA clustering method (Fig. 4) visualizes the genetic relationships among the 83 accessions based on SSR data. While some accessions with similar classifications do appear in proximity, the tree does not show strict separation between hemp and marijuana samples. Instead, accessions from both types are distributed across multiple branches, reflecting shared ancestry and partial clustering rather than clearly isolated lineages. Notably, several marijuana samples appear interspersed within broader hemp-associated

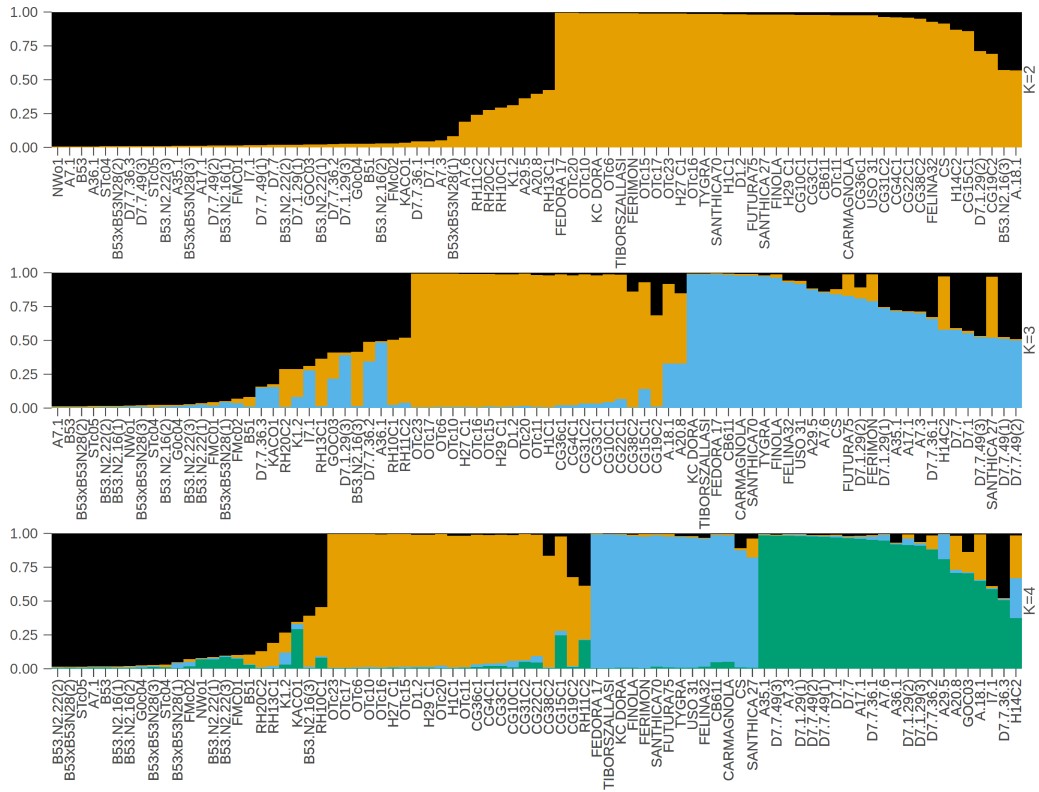

**Figure 1  STRUCTURE bar plot for $K = 2$, $K = 3$, and $K = 4$, with each bar representing an individual sample.** The most likely number of clusters (K) identified is $K = 4$. The samples are displayed based on STRUCTURE-assigned clusters and reflect the underlying genetic structure. The clusters when viewed from left to right are: Cluster A (black color, European fiber-type hemp), Cluster B (orange color, varied collection of accessions with different chemotype backgrounds), Cluster C (blue color, marijuana accessions), Cluster D (green color, high-THC cultivars).

clades. The color-coding in Fig. 4 (black for hemp, red for marijuana) was applied post hoc, based on predefined chemotype classifications, to facilitate visual comparison and does not originate from the phylogenetic algorithm itself.

The four-group model was also supported by the UPGMA dendrogram based on SSR genetic distances and is further associated with the clusters as shown in STRUCTURE analysis.

## Comperative assessment of STRUCTURE, PCoA, and UPGMA

To sum up, we cross-referenced the genetic clusters with the phylogenetic groupings from the UPGMA dendrogram and the coordinate patterns from PCoA to evaluate the robustness of the population structure inferred by STRUCTURE at $K = 4$. The four STRUCTURE clusters, which are referred to as Clusters A through D in this context, demonstrated a high degree of consistency throughout the three analyses.

The comparative assessment of STRUCTURE, PCoA, and UPGMA analyses consistently indicates the existence of four genetically different groups within the dataset. All three analytical approaches show these categories clearly, and they also show that the sampled

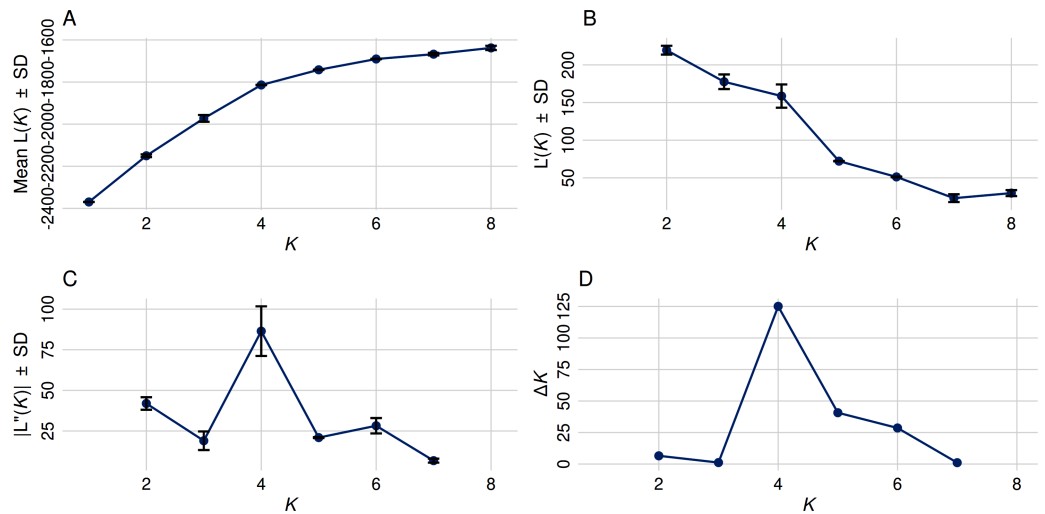

**Figure 2  Optimal cluster determination by the Evanno method.** (A) The Mean Ln P(K) ± Stdev method suggested that $K = 5$ provides the best fit, as Ln P(K) continued to increase through $K = 8$. (B) First-order rate of change in Ln P(K) [L'(K)] ± Stdev between successive K values. (C) Second-order rate of change in Ln P(K) [L'(K)] ± Stdev between successive K values. (D) The $\Delta$ K method detected the highest rate of change at $K = 4$, indicating four distinct genetic subpopulations.

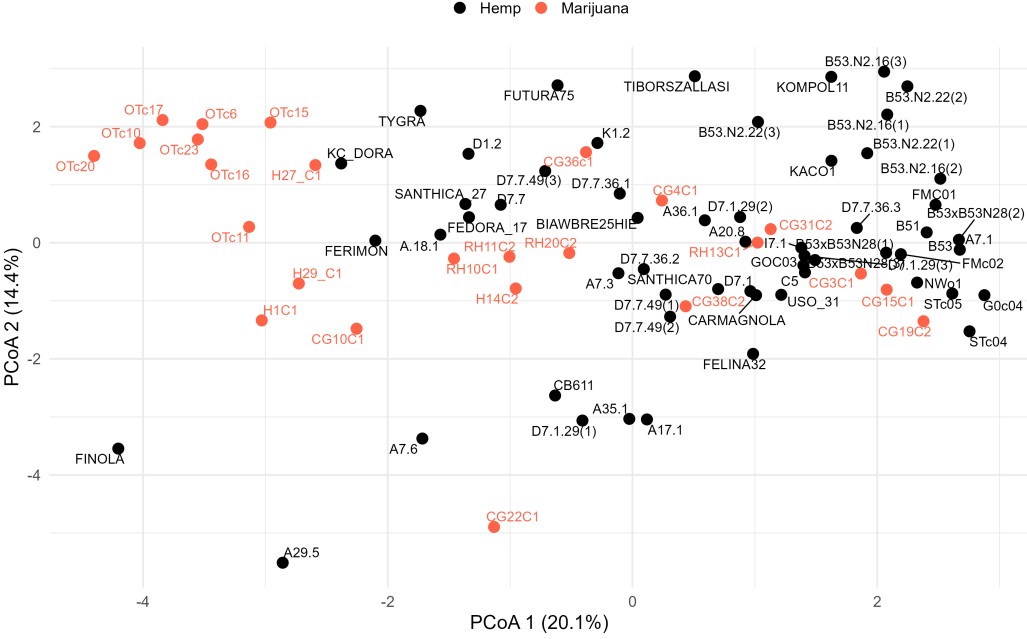

**Figure 3  Principal coordinate analysis (PCoA) of the cannabis genotypes.** The plot shows the first two principal coordinates (PCoA 1 and PCoA2), which account for 20.1% and 14.4% of the total variance, respectively. Hemp samples are shown with black dots and marijuana samples are shown with red-orange ones.

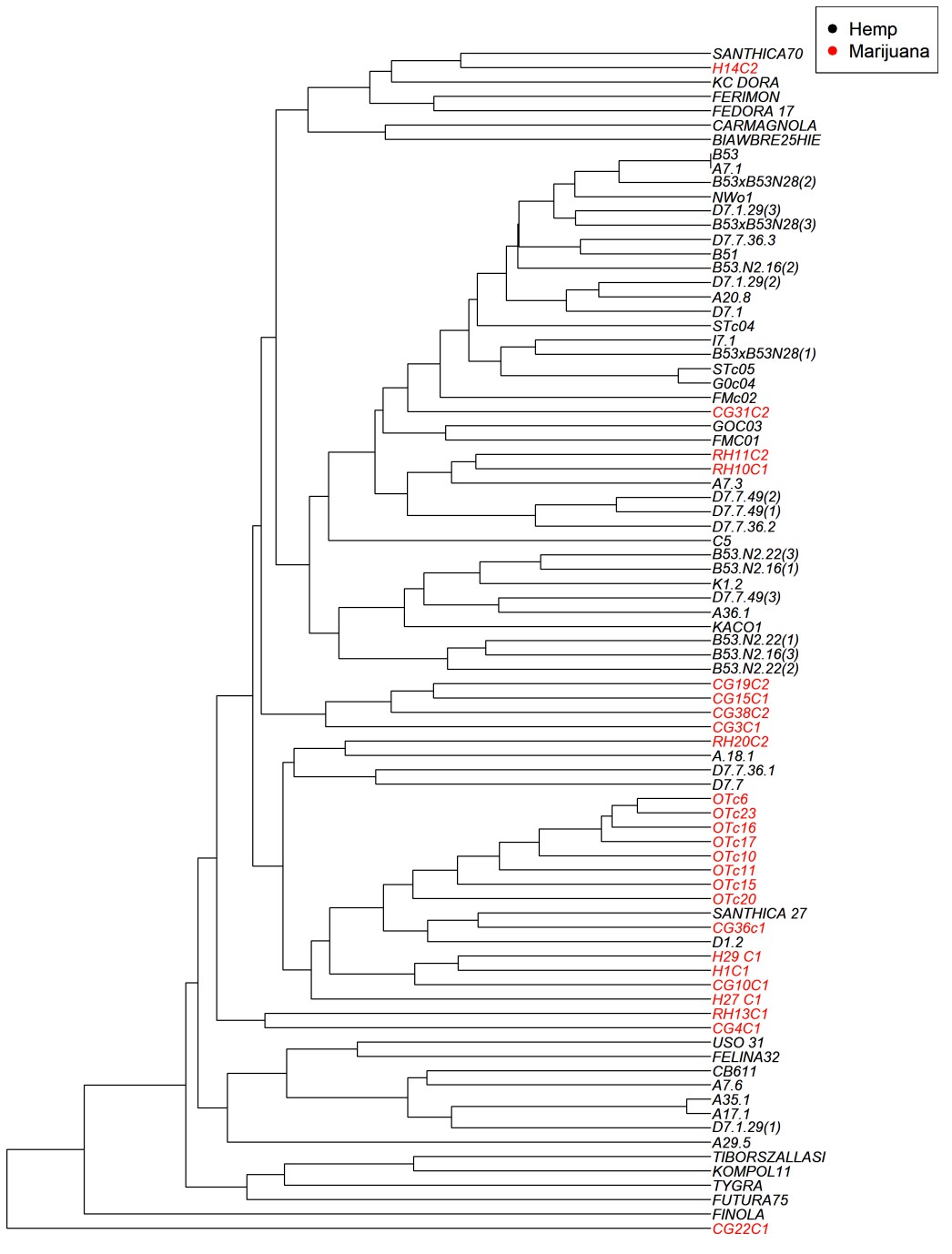

**Figure 4 UPGMA dendrogram illustrates the genetic relationships among the cannabis genotypes.**
The branches are color-coded to indicate whether the sample is hemp or marijuana. The "Hemp" type branches are shown in black and the "Marijuana" type branches are shown in red.

cannabis accessions have coherent genetic links, which strengthens the estimated population structure.

In the PCoA, the first group is clearly different from the others. It is in the lower-left quadrant and shows a lot of separation along both PCoA axes. The same group forms a strong monophyletic clade in the UPGMA dendrogram and shows consistent ancestry in the STRUCTURE analysis. Some important examples in this category are FINOLA, FUTURA75, TIBORSZALLASI, KOMPOL11, TYGRA, CG22C1, A29.5, A7.6, and CB611. Most of these accessions are industrial hemp cultivars that are known for fiber production and minimal levels of THC. Their tight genetic clustering suggests a high degree of genetic conservation, possibly due to selective breeding for similar agronomic traits.

The second group, which is mostly made up of drug-type cannabis samples, is in the upper-left corner of the PCoA diagram. In the UPGMA tree, these samples make up a single subclade, and they are also all part of the same STRUCTURE ancestry component. This group has the following: CG10C1, CG15C1, CG19C2, CG31C2, CG36C1, CG38C2, H1C1, H14C2, H27 C1, H29 C1, OTc10, OTc11, OTc15, OTc16, OTc17, OTc20, OTc23, and OTc6. These accessions are genetically different from the industrial hemp cluster and are high-THC types. They were probably chosen for their production of psychoactive compounds rather than for fiber-related traits.

In all three analyses, the third group displays a mixed or intermediate pattern. In the PCoA, these samples are located across groups that are genetically more different, which suggests that they share a common ancestor. The STRUCTURE analysis shows a mix of genetic contributions, indicating gene flow between hemp and marijuana types. Samples in this category include RH10C1, RH11C2, RH13C1, RH20C2, GOC03, FMC01, FMc02, and G0c04, to name a few. The UPGMA clustering pattern supports this observation, showing that these accessions form a moderately supported subclade that connects the two main clades. This group could be hybrids or transitional cultivars that have mixed genes because of historical crossover between different gene pools.

The fourth group, which contains FEDORA 17, FERIMON, KC DORA, SANTHICA70, CARMAGNOLA, USO 31, and others, is in the upper-central part of the PCoA plot and makes up a separate clade in the UPGMA tree. STRUCTURE also groups most of these samples based on their ancestry. Most of the plants in this group are European fiber hemp variants. They are genetically different from both the high-THC group and the industrial hemp group (Group 1) previously described. The genetic diversity revealed here likely reflects a distinct breeding lineage and selection history tailored to fiber or seed production within European contexts.

Cross-validation using three different analytical techniques provides validity to the idea that these clusters represent biologically meaningful subpopulations. Furthermore, it supports the idea that the observed genetic structure is not just a product of regulatory or chemotypic classification, but instead reflects historical divergence and breeding lineage. Rather than a simple hemp-marijuana binary, the consistent clustering across STRUCTURE, PCoA, and UPGMA supports a population structure influenced by diverse breeding trajectories and validates the existence of four genetically distinct groups.

**Table 3** Outline of: (A) The AMOVA framework for the 83 samples. (B) F-statistics values (Fst, Fis, Fit). (C) Allelic patterns for codominant data for marijuana and hemp type (mean and standard error values). (D) Nei's genetic identity and distance for Marijuana and Hemp type.

### AMOVA Table

| Source | df | SS | MS | Est. Var. | % |
|---|---|---|---|---|---|
| Among populations | 1 | 29.613 | 29.613 | 0.354 | 9% |
| Among individuals | 81 | 392.862 | 4.850 | 1.428 | 38% |
| Within individuals | 83 | 165.500 | 1.994 | 1.994 | 53% |
| Total | 165 | 587,976 | | 3.776 | 100% |

| F-Statistics | Value | P(rand >= data) |
|---|---|---|
| Fst | 0.094 | 0.001 |
| Fis | 0.417 | 0.001 |
| Fit | 0.472 | 0.001 |

### Allelic Patterns for Codominant Data

| Mean values | | | Standard Error (SE) values | | |
|---|---|---|---|---|---|
| Population | Marijuana | Hemp | Population | Marijuana | Hemp |
| Na | 5.000 | 8.000 | Na | 0.577 | 1.390 |
| Na Freq. >= 5% | 3.900 | 3.900 | Na Freq. >= 5% | 0.407 | 0.407 |
| Ne | 3.117 | 3.473 | Ne | 0.469 | 0.403 |
| I | 1.205 | 1.415 | I | 0.144 | 0.161 |
| No. Private Alleles | 0.800 | 3.800 | No. Private Alleles | 0.359 | 1.052 |
| No. LComm Alleles (<=25%) | 0.000 | 0.000 | No. LComm Alleles (<=25%) | 0.000 | 0.000 |
| No. LComm Alleles (<=50%) | 0.000 | 0.000 | No. LComm Alleles (<=50%) | 0.000 | 0.000 |
| He | 0.603 | 0.662 | He | 0.060 | 0.050 |
| uHe | 0.616 | 0.668 | uHe | 0.062 | 0.051 |

| Population | Nei's genetic identity | Population | Nei's genetic distance |
|---|---|---|---|
| Marijuana | 0.750 | Marijuana | 0.288 |
| Hemp | 0.250 | Hemp | 0.712 |

## Analysis of molecular variance

Analysis of molecular variance (AMOVA) was performed based on codominant genotypic distances and the results (Table 3A) revealed the genetic structuring between hemp and marijuana accessions. Of the total genetic variation, 9% was attributed to differences between hemp and marijuana groups, 38% to differences among individuals within groups, and 53% to within-individual variation. The fixation index ($F_{st} = 0.094$, $P = 0.001$, Table 3B) indicates moderate genetic differentiation between the two populations. This hierarchical structure reflects both historical divergence and domestication pathways shaped by selection pressures on hemp and marijuana accessions. These results suggest a moderate but distinct genetic structure in *C. sativa*, with partial separation between hemp and marijuana.

Allelic diversity metrics (Table 3C) were compared between marijuana and hemp groups. Hemp had a higher mean number of alleles (Na = 8.0) than marijuana (Na = 5.0), indicating greater allelic richness. The effective number of alleles (Ne) was also greater in

hemp (3.473) than in marijuana (3.117), further supporting higher genetic diversity in hemp. Private alleles were more common in hemp (mean = 3.8) compared to marijuana (mean = 0.8). Expected heterozygosity (He) and unbiased He (uHe) were both higher in hemp (0.662 and 0.668, respectively) than in marijuana (0.603 and 0.616, respectively). Genetic divergence between groups was assessed using Nei's genetic distance and identity (Table 3D). Nei's genetic distance between hemp and marijuana was 0.288, with a genetic identity of 0.750, indicating moderate differentiation. Nei's genetic identity (0.750) and genetic distance (0.288) reflect moderate similarity and divergence between hemp and marijuana groups, indicating distinct allele frequencies between these populations.

### High-resolution melting (HRM) assay

The two sex-specific SNPs that were detected were homozygous in all females and heterozygous in all males (*Gilchrist et al., 2021*). Thus, based on differences in fluorescence signals at certain melting temperatures (Tm), the optimal temperature range for heteroduplex melting was found to be from 70 °C to 80 °C and the female and male samples had different melting curves (Fig. 5). Female homozygous samples are depicted with red lines (variant1) and male samples with blue lines, (variant2), showing difference in melting behavior. Specifically, the melting curves from homozygous variant samples have a relatively similar form and can be distinguished from one another by variations in Tm. Heterozygotes produce distinct melting curve profiles compared to homozygotes due to the formation of heteroduplexes, which alter the shape of the curve (*Liew et al., 2004*; *Druml & Cichna-Markl, 2014*). Using HRM analysis, 73 female and 10 male plants were identified, and were consistent with phenotypic observations (Table 1).

## DISCUSSION

Given the ongoing interest in the applications and economic potential of *C. sativa*, this study contributes to the growing body of work on its genetic structure by assessing variability across 83 accessions using SSR markers. Herein, genetic diversity among 83 *C. sativa* genotypes was assessed using 10 simple sequence repeat (SSR) markers, which are commonly used in *C. sativa* genetic research due to their codominant inheritance, high reproducibility, multiallelic nature, and widespread genomic distribution (*Madesis, Ganopoulos & Tsaftaris, 2013*; *Borin et al., 2021*). These markers have been effectively applied in previous studies to evaluate genetic variability (*Soler et al., 2017*; *Ioannidis et al., 2022*; *Adamek et al., 2023*; *Pandey et al., 2023*; *Benkirane et al., 2024*) and distinguish between fiber- and drug-type cultivars (*Borin et al., 2021*). Preliminary evaluation of regionally adapted landraces using SSR markers can inform parent selection and guide breeding strategies (*Shams et al., 2020*).

### Genetic population structure in *Cannabis sativa*

According to this study, the genetic structure of *C. sativa* is more complicated than the widely accepted binary or chemotype-based classifications (such as Types I–V or 'hemp *vs.* marijuana'). We applied STRUCTURE analysis at $K = 4$ to identify four genetically distinct clusters. The PCoA and UPGMA dendrogram results consistently supported these

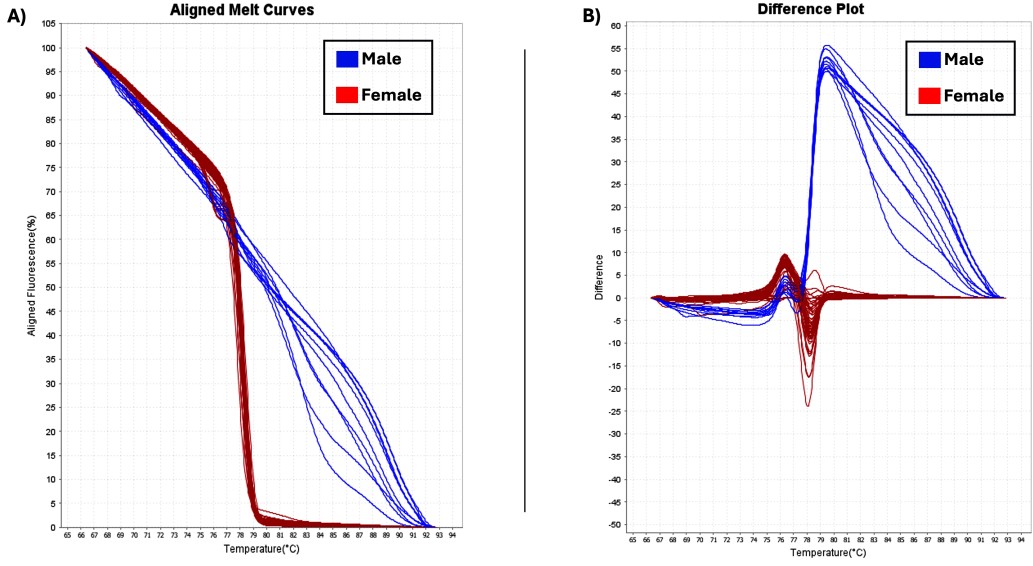

**Figure 5  HRM analysis.** (A) Aligned melt curves. Data were collected during an HRM curve experiment and present the melt regions. The curve colors represent the 83 *C. sativa* samples. General HRM profiles, based on Tm, revealed two different variants: variant1 (red line) and variant2 (blue line) which represent the female and male samples, respectively. (B) Difference plot. By using 83 samples, two distinct categories were revealed.

clusters, confirming the existence of a population substructure influenced by a range of breeding histories, geographic origins, and selection pressures. These patterns appeared without regard to chemotype classifications, highlighting the drawbacks of relying solely on cannabinoid profiles as a stand-in for genetic identity (*Sawler et al., 2015*; *Soler et al., 2017*).

Each of the four STRUCTURE clusters represents a distinct genetic lineage within *C. sativa*, according to their own interpretation. A cohesive gene pool that was probably influenced by industrial hemp breeding was reflected by the close clustering of these accessions in both PCoA (left quadrant) and UPGMA (lower branches). The closely clustered samples exhibited genetic similarities (*Ullah et al., 2023*). With its more admixed or intermediate accessions, Cluster B covered central regions in both STRUCTURE and PCoA and showed up as widely dispersed branches in UPGMA, which may indicate hybrid origins or introgressed backgrounds. Cluster C exhibited tight grouping in the upper right of the PCoA and formed a central subtree in the UPGMA dendrogram, primarily consisting of marijuana accessions (*e.g.*, CG31C2, FMc01, CG10C1). Cluster D was highly differentiated, occupying an upper-right clade in UPGMA and a distinct quadrant in PCoA. It contained high-THC cultivars such as D7.7.49(2) and D7.7.36.2. According to these results, chemotype data (such as THC *vs.* CBD content) offer useful phenotypic context, but they fall short of capturing the entire range of genetic diversity and evolutionary divergence in *C. sativa* (*Sawler et al., 2015*). Human-mediated selection and domestication history have shaped the neutral population structure reflected by the SSR-based clusters, which may or may not correspond with chemotype. Specifically, the
substructure found in hemp, covering at least three different genetic clusters, indicates the presence of several breeding pools, perhaps from various geographical locations or chosen for different agronomic characteristics like fiber, grain, or CBD production (*Soler et al., 2017*).

Although chemotype-based classifications are helpful in characterizing the amount of cannabinoid content, they may not accurately represent the underlying genetic structure. SSR markers show population divergence influenced by past lineage, domestication events, or breeding programs because they are neutral and non-coding. Regardless of the chemotype designation, our results indicate that there are four low-admixture clusters among the 83 genotypes. This implies that the actual structure of the germplasm collection is more complex than a straightforward classification based on THC/CBD (*Sawler et al., 2015*; *Pan et al., 2021*).

Although it is not possible to assume causality, the alignment of certain chemovars with genetic clusters probably reflects indirect effects of selection or shared breeding ancestry. This inference is supported by prior studies showing that population structure often aligns with chemotype variation due to breeding histories, but not necessarily because of direct genetic linkage (*Sawler et al., 2015*; *Vergara et al., 2019*). To resolve functional variation patterns in relation to chemotype, future research utilizing genome-wide SNP data would be required. Even though this study did not directly assess genetic rarity, the presence of unique alleles and moderate population structure suggests that focused breeding and conservation strategies may help maintain broader genetic diversity in *C. sativa* (*Alsaleh & Yılmaz, 2025*).

## Genetic analysis of individuals, *Cannabis'* population and phylogenetic relationships

All fixation indices (Fst, Fis, Fit) were statistically significant ($P = 0.001$), indicating meaningful genetic structure in the dataset. The observed Fst of 0.094 reflects moderate differentiation between the hemp and marijuana groups, consistent with prior studies in *C. sativa* (*Sawler et al., 2015*; *Schwabe & McGlaughlin, 2019*). This supports a model of shared ancestry with divergence driven by selective breeding and gene flow. The elevated Fis (0.417) suggests a heterozygote deficit within groups, potentially due to non-random mating or inbreeding within breeding programs (*Wright, 1951*; *Borin et al., 2021*). The overall Fit value of 0.472 points to structured inbreeding across the full set of accessions. Together, these results reflect moderate genetic partitioning and directional selection, aligning with previous SSR- and SNP-based studies that reported population structure shaped by domestication history and trait selection (*Gao et al., 2014*; *Barcaccia et al., 2020*).

Nei's genetic distance between hemp and marijuana was estimated at $D = 0.288$, with a corresponding genetic identity of $I = 0.750$. This indicated moderate genome-wide differentiation between the two groups, which probably reflects divergent selection during domestication, for high cannabinoid production in marijuana and for fiber-related traits in hemp. These results are consistent with previous reports of broad genetic clustering aligned with functional use (*Sawler et al., 2015*; *Ren et al., 2021*). Although both groups exhibit considerable shared ancestry, the preservation of distinct germplasm pools remains

critical to safeguarding the unique allelic diversity characteristic of each lineage (*Barcaccia et al., 2020*).

## Analysis of molecular variance

Analysis of molecular variance (AMOVA) revealed that the total molecular variance was divided into three components: among populations (9%), among individuals (38%) and within individuals (53%). Most of the variation in our study was detected, based on AMOVA, within individuals, pointing out the inherent genetic variety of each of the genotypes, which according to *Benkirane et al. (2024)* is due to heterozygosity.

Allelic patterns across the two groups (marijuana and hemp) were analyzed. The mean number of alleles (Na) was higher in hemp samples, indicating a greater allelic count compared to marijuana. Similarly, the number of effective alleles (Ne) was also higher in hemp, suggesting a higher level of genetic diversity and a broader, more evenly distributed genetic base. This greater allelic and effective allelic diversity in hemp likely reflects its long history of cultivation for various industrial purposes, such as the production of fiber, grain and seed oil. Each of these purposes requires distinct traits, leading to the preservation of a broader gene pool through diverse selection pressures (*Gao et al., 2014*; *Sawler et al., 2015*). In contrast, marijuana has undergone more intense and consistent selection for a narrow set of traits, such as high cannabinoid content, especially THC, over the past few decades (*Clarke & Merlin, 2014*). The hybridization of cultivated varieties and human-mediated seed transport may have contributed to a reduction in genetic diversity. Thus, the existence of distinct genetic groups and their unique profiles may reflect some level of genetic narrowing due to selective breeding practices (*Lynch et al., 2016*; *Vergara et al., 2019*).

Regarding private alleles, hemp accessions exhibited a mean of 3.8 private alleles per group, significantly more than the 0.8 observed in marijuana. These alleles are valuable for varietal registration, traceability, and identification, especially in regulatory contexts where distinguishing between hemp and marijuana cultivars is crucial, in addition to their role in assessing genetic diversity (*Borin et al., 2021*).

Regarding the expected heterozygosity (He) and unbiased expected heterozygosity (uHe), their values were slightly higher in hemp than marijuana. *Soler et al. (2017)*, who studied seed lots, used six gSSR markers to interpret the genetic structure of 18 cultivars of *C. sativa* var. *indica* (marijuana accessions) and two *C. sativa* var. *sativa* (hemp accessions). They reported variations in He and stated that the higher observed and predicted heterozygosity values were most likely brought on by increased inbreeding, which increases homozygosity and decreases diversity in small populations through genetic drift.

## Sex determination

Gender identification with gender-specific markers is vital to optimize breeding efforts in cannabis, by facilitating the identification and selection of plants (*Alsaleh & Yılmaz, 2025*). Because plant gender influences selection strategies and economic value, breeding programs benefit from separating male and female plants at the seedling stage (*Borin et al., 2021*). Here we identified 73 female and 10 male plants, a result that matched the reported

phenotypes and helped us proceed further with the selection of THC rich female plants (*Techen et al., 2010*).

Notably, several markers for sex identification in cannabis have been published, such as MADC (Male-Associated DNA from *C. sativa*) markers. The first one that was designed was MADC1 (*Sakamoto et al., 1995*), followed by MADC2 (*Mandolino et al., 1999*), MADC3, MADC4 (*Sakamoto et al., 2005*), MADC5 and MADC6 (*Törjék et al., 2002*). These RAPD (random amplified polymorphic DNA) markers are dominant and suffer from low reproducibility within and between laboratories, which limits their reliability and scalability for high-throughput applications (*Babu et al., 2020*). Later, *Toth et al. (2020)* developed the CSP-1 assay which predicted sex. *Pan et al. (2021)* designed two InDel primers, Cs-I1–10 and Cs-I1–15, which demonstrated broader applicability than those previously reported. Also, *Torres et al. (2022)* detected MADC2 by using high-throughput qPCR achieving 98.5% accuracy. In this research, by using the SNP markers created by *Gilchrist et al. (2021)*, the detection of male and female plants with the HRM analysis was rapid, inexpensive, and accurate. However, a key limitation of the HRM-based sex identification approach used in this study is its inability to detect phenotypic variations, like hermaphroditism in female plants. These phenomena can occur in *C. sativa* due to environmental stress, genetic predisposition, somatic mutations, or age-related processes. Consequently, while the HRM assay provides a reliable and efficient method for early genetic sex determination, its results should be interpreted with caution, especially in breeding programs (*Toth et al., 2020*; *Torres et al., 2022*).

## CONCLUSIONS

To conclude, this study contributes to understanding the genetic diversity and sex identification of *C. sativa* using microsatellite markers and HRM analysis. While not a genomic-scale study, the findings support the presence of structured diversity between hemp and marijuana groups. Most importantly, the use of HRM for sex identification offers a practical, rapid, and cost-effective method that can be implemented in-house with the appropriate equipment. Although no sex test is 100% accurate, this approach provides a useful tool for early screening, which can help reduce the allocation of resources to unwanted male plants and improve breeding efficiency.

### Funding
The authors received no funding for this work.

### Competing Interests
Ioannis Ganopoulos is an Academic Editor for PeerJ. Panagiotis Karnoutsos and Marios Karagiovanidis are employed by IA Agro S.M.P.C. and Dimitrios Magalios and Christos Nanos are employed by IG Agrotech P.C. Vangelis Mitsis is employed by Ekati Alchemy Lab SL.

## Author Contributions

- Anastasia Boutsika conceived and designed the experiments, performed the experiments, analyzed the data, prepared figures and/or tables, authored or reviewed drafts of the article, and approved the final draft.
- Eleftheria Deligiannidou conceived and designed the experiments, performed the experiments, prepared figures and/or tables, authored or reviewed drafts of the article, and approved the final draft.
- Theodoros Moysiadis analyzed the data, prepared figures and/or tables, and approved the final draft.
- Nikolaos Tourvas performed the experiments, analyzed the data, prepared figures and/or tables, authored or reviewed drafts of the article, and approved the final draft.
- Panagiotis Karnoutsos analyzed the data, prepared figures and/or tables, authored or reviewed drafts of the article, contributed reagents, and approved the final draft.
- Marios Karagiovanidis analyzed the data, prepared figures and/or tables, authored or reviewed drafts of the article, contributed reagents, and approved the final draft.
- Dimitrios Magalios performed the experiments, prepared figures and/or tables, authored or reviewed drafts of the article, and approved the final draft.
- Christos Nanos performed the experiments, prepared figures and/or tables, authored or reviewed drafts of the article, and approved the final draft.
- Vangelis Mitsis performed the experiments, prepared figures and/or tables, authored or reviewed drafts of the article, contributed reagents, and approved the final draft.
- Eleni Tsaliki conceived and designed the experiments, prepared figures and/or tables, authored or reviewed drafts of the article, and approved the final draft.
- Eirini Sarrou conceived and designed the experiments, prepared figures and/or tables, and approved the final draft.
- Apostolos Kalivas conceived and designed the experiments, prepared figures and/or tables, and approved the final draft.
- Ioannis Ganopoulos conceived and designed the experiments, prepared figures and/or tables, authored or reviewed drafts of the article, and approved the final draft.

## Data Availability

The raw measurements are available in the Supplementary File.

## Supplemental Information

Supplemental information for this article can be found online at http://dx.doi.org/10.7717/peerj.19770#supplemental-information.

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
