# Peer review of "Molecular profiling and sex determination of Cannabis sativa germplasm collection: Exploring microsatellite markers and high-resolution melting (HRM) analysis"

_PeerJ, doi:10.7717/peerj.19770_

## Round 0.1 · original submission · Major Revisions

· Academic Editor

Major Revisions

Dear colleagues, the reviewers have now concluded their assessment of your manuscript, and based on the reports your work has merit for publication, following a revision. Please also provide a rebuttal letter alongside your revised manuscript with a point-to-point response to the reviewers' comments

Reviewer 1 ·

Basic reporting

Figures
The figures are poor quality and very hard to see.

References
I appreciate the references included in the manuscript, as they highlight important foundational studies. However, many of the references cited are somewhat outdated. Incorporating more recent studies would strengthen the manuscript by reflecting the latest advancements in Cannabis genetics and taxonomy. For example:
1. Taxonomy and Classification
o Lapierre, É., Monthony, A.S., & Torkamaneh, D. (2023). Genomics-based taxonomy to clarify cannabis classification. Genome, 66(8), 202-211.
2. Geographic Origins
o Ren, G., Zhang, X., Li, Y., et al. (2021). Large-scale whole-genome resequencing unravels the domestication history of Cannabis sativa. Science Advances, 7(29), eabg2286.
o Osterberger, E., Lohwasser, U., Jovanovic, D., et al. (2022). The origin of the genus Cannabis. Genetic Resources and Crop Evolution, 69(4), 1439-1449.
3. Population Structure and Genetic Diversity
o Zhang, J., et al. (2020). Genetic diversity and population structure of cannabis based on the genome-wide development of simple sequence repeat markers. Frontiers in Genetics, 11, 958.
o Henry, P., et al. (2020). A single nucleotide polymorphism assay sheds light on the extent and distribution of genetic diversity in cultivated North American cannabis. Journal of Cannabis Research, 2, 1-11.
Integrating these references, along with others relevant to your topic, will provide a more comprehensive and updated context for your research. I have listed others in my specific comments below.

Type of Cannabis Samples
Providing details about the type of Cannabis (e.g., hemp, marijuana, Type 1/2/3) is crucial, as these groups are known to exhibit significant genetic divergence. This context is essential for interpreting your results.

Marker Information
• Table 2 lacks key information, such as the microsatellite sequences and the number of repeating units. This is vital for reproducibility and understanding marker development.

Experimental design

Methods and Results
The results and subsequent analyses rely heavily on accurate fragment data. Unfortunately, the current method for calling or binning the data appears to have issues.

In ANUCS303, the reported range (142–157) does not align with the table data (124–209). Additionally, peaks at 140 and 141 suggest a mononucleotide SSR, which is highly unusual given that SSRs are typically 2–6 nucleotide units.

B05-CANN1 exhibits irregular peak patterns inconsistent with a trinucleotide SSR, where peaks should reflect changes in 3-nucleotide increments (e.g., 205, 208, 211).
Please revisit the fragment data processing, ensure proper binning, and clarify the software or methodology used for these calculations.

Population Genetic Analyses
• The STRUCTURE analysis suggests a single genetic group rather than four, as the genetic assignments are not distinct. This might stem from data processing issues or insufficient variation in the markers.
• The PCA results also lack clear clustering. Addressing the fragment data concerns could rectify this.

Validity of the findings

The findings presented in this study have potential significance, but there are areas that require attention to fully establish their robustness and validity. Below are key observations:

Data Provision and Statistical Soundness
While the authors have included substantial data, there are concerns regarding the integrity of the fragment data and its binning process. These issues impact the robustness of the statistical analyses and the conclusions drawn. Revisiting and correcting these foundational data processing steps is essential to ensure statistical soundness and reproducibility. Once addressed, this could strengthen the study's reliability.

Conclusions and Alignment with Research Questions
The conclusions presented are generally linked to the original research questions; however, they are significantly impacted by the aforementioned data issues. For example, claims about population clustering and genetic diversity are based on results that lack sufficient support due to incorrect data binning. As such, conclusions should be re-evaluated once the data are properly processed and analyzed.

Replication and Contribution to Literature
This study includes important analyses such as PCA and STRUCTURE, which are valuable for understanding population structure. However, the rationale for replication or advancing previous findings should be better articulated. Clarifying how this work adds value to the existing literature—beyond replicating standard methods—would strengthen the manuscript's novelty and impact.

Novelty and Contribution to Field
While the impact and novelty of the study are not formally assessed, addressing the current data concerns will allow a clearer evaluation of the study's contribution to the field. This could be particularly meaningful if the authors highlight unique aspects of their markers or dataset, such as new insights into Cannabis genetic diversity or novel applications of their methodology.

Additional comments

Specific Revisions, comments, and suggestions are in the attached document.

Annotated reviews are not available for download in order to protect the identity of reviewers who chose to remain anonymous.

·

Basic reporting

The manuscript is written in correct English using technical vocabulary in the correct manner.
Literature references are well adopted and fit with the cited topic, although some more references should be provided in specific sections, latter discussed.
Manuscript structure and Tables are well organised, although some in-section modification could be made, but Figured should be provided in better quality as the text is difficult to read. Also, some revisions have to be made in the legends of some figures, as later discussed.
Manuscript is self-contained and the results are obtained in reliable ways, but some revisions are expected before acceptance for publication.



Line 46. PCA indicates the Principal Component Analysis, not “Coordinate”, please revise

Line 75. It seems the word “genus” is incorrect as it is used before the species name

Lines 103-105. This sentence is lacking a comma, which makes it difficult to understand

Lines 122-123. “The collected young leaf tissues from 83 plants were immediately frozen in liquid nitrogen and stored at -80°C until DNA extraction.” Should be placed before mentioning the extraction kit used for DNA extraction to better fit the sampling and DNA extraction workflow description.
Line 124. Please provide the spectrophotometer brand and model used for quantification and quality evaluation of the gDNA extracted

Line 128. Please provide references from which the SSR markers were used in the present study were selected.

Line 132. Please indicate the PCR Master Mix supplier and concentration. Also, add the primers molarity (M) used and the genomic DNA concentration adopted

Lines 153-154. Please add the molarity (M) used for the primers and correct “DNTPs”, which should be written “dNTPs”. Also, provide the Taq polymerase manufacturer
Line 158. Please provide the manufacturer of the QuantStudio 5.
E.g. QuantStudio 5 (Waltham, MA, US). Please do the same for the rest of the kits, reagents and instruments used

Line 165. Please provide, also in the same field, al the available citations for the R package adopted
Line 183. Please explain why the Silhouette is reported to indicate the best number of clusters as K=4 and not K=5 as this last is the higher point in the graph in Fig. 1B
Lines 193-195. Please provide an explanation for using accession STc04 as an outgroup. Moreover, it seems that there is not accordance between what is written in the manuscript and in the Figure legend: “less diverse branches in purple and more diverse branches in teal”. Also, please highlight the identified clusters in the UPGMA to make them easier to identify
Lines 199-203. Please, provide here or in the M&M section the parameters used for the STRUCTURE and fastSTRUCTURE analyses, and explain why Fig 1I should indicate K=4 and not K=5.

Line 237. Please, provide the “certain” melting temperatures in the text

Discussion: The Discussion section mostly repeats the Results one but providing slightly enhanced highlights. For this reason, it may be better to reorganise it and to use additional references to justify the main results obtained from the analyses. Moreover, some discussed results may be misleading, like Fst below 0.15, that indicates vary poor discriminability among the populations/clusters identified. Moreover, PCA shows vast overlapping among clusters, which makes them difficult to define as distinguishable. Also, the K=4 STRUCTURE plot shows that all the analysed accessions present the same or comparable memberships with the ancestral groups identified, and Evanno plot for DeltaK values is limited to 5 (not presenting any data) and may be considered to increase the iterations and putative Ks for this analysis, thus not limiting it to 5, but alt least to 10 with MCMC 100’000 and burn-in 20000.
Noteworthy, the majority of the Molecular Variance is represented within individuals (more than 60%) while only 15% is represented among the identified clusters.

I suggest the authors to revise the analytical methodology adopted on the obtained dataset

Experimental design

The considered number of samples can be adequate and representative of the genetic variance observable in the species. This said, the statistical approaches adopted to interpret the genotyping results lacks precision and should be revised or justified with further information from the authors. Despite this, the sex-determination method proposed in the present study appears to be reliable, precise, and efficient, although a higher number of samples would provide further information on its precision and reliability, maybe considering more populations from different germplasm collections.

Validity of the findings

Due to the lack of information provided on the parameters used for the genetic statistics anc clustering analyses, the manuscript presents interesting findings which should be better reported and discussed to confirm their reliability.

Additional comments

Despite the issues identified and previously discussed, the manuscript presents interesting findings, particularly regarding the proposed sex-determination approach. However, major revisions are necessary to make the manuscript suitable for publication. Specifically, the authors should provide detailed information on the methodology used to obtain and analyze the genotypic data and enhance the discussion section with more precise comments on the findings and results. Additionally, the manuscript's structure should be revised and improved to ensure a logical presentation of the Methods, Results, and Discussion sections.

---

## Round 0.2 · Major Revisions

· Academic Editor

Major Revisions

Dear colleagues, the reviewers have both delivered their reassessment of your work and evidently, there are still some issues, mainly with the binning of SSRs and relevant analyses. Based on these comments I must recommend a major revision

Reviewer 1 ·

Basic reporting

At this stage, the manuscript does not fully meet the standards of Basic Reporting. While the manuscript is written in English and generally maintains a professional tone, there are several areas that require significant attention:

Clarity and Consistency: There are still unresolved issues related to the methods and results that affect the overall clarity of the manuscript. Specifically, inconsistencies in microsatellite binning, ambiguous data processing methods, and lack of clarity about the software used undermine confidence in the results presented.

Literature Context: Although the manuscript includes relevant citations, it would benefit from a more robust discussion of methodological choices, particularly with respect to data analysis (e.g., PCA vs. PCoA), and how these choices are supported by population genetic literature.

Structure and Figures: The article structure generally conforms to professional standards, but several figures (e.g., PCA/PCoA, dendrogram, STRUCTURE plot) are difficult to interpret and not well-explained. Figure legends and axes should be clarified, and sample ordering within STRUCTURE plots should follow a logical and informative arrangement. It is also unclear whether Figure 1 represents a PCA or PCoA; this inconsistency should be corrected.

Data Availability: Although the authors state that they have corrected fragment binning and rerun analyses, the raw data and updated methodological parameters (e.g., binning thresholds, distance matrices) have not been clearly provided or explained. Transparency in this area is essential for reproducibility.

Self-Containment and Relevance: The manuscript attempts to address population structure and variety classification, but these results are currently undermined by unresolved issues in data processing. Until the data are re-evaluated and clearly presented, it is difficult to assess whether the results meaningfully address the hypotheses.

Given these outstanding concerns, I recommend a major revision focused on the integrity and transparency of the data and analyses before further consideration.

Once the foundational data issues are fully resolved and the methods/results are accurately aligned, I would be happy to provide a more detailed review of the manuscript’s structure and presentation.

Experimental design

While the manuscript falls within the scope of the journal and presents original research on genetic diversity and sex determination in Cannabis sativa, several critical aspects of the experimental design require clarification or revision:

The manuscript would benefit from a clearer articulation of the research question and the specific knowledge gap it addresses. Although genetic characterization of cannabis is a relevant topic, the rationale for focusing on “breeding line” vs. “registered variety” as the primary comparison groups is not well justified. It is unclear whether these categories reflect meaningful biological or historical differentiation, especially when compared to more established classification systems such as Type I–IV or drug-type vs. hemp-type. A more robust explanation of how this classification aligns with the study goals is needed.

There are persistent issues with allele binning in the SSR data. Several loci display allele lengths that do not align with expected repeat unit sizes (e.g., not conforming to multiples of 2 or 3 bp), which compromises the accuracy of allele calling. Correct binning is essential for valid downstream analyses (STRUCTURE, PCoA, AMOVA). The manuscript should include explicit details about the software and settings used for fragment binning, as well as a table showing corrected allele bins for each marker.

Although general software tools are listed (e.g., GeneMapper, GenAlEx, R packages), the exact parameters used for key analyses—such as distance matrix construction for PCoA, binning tolerances for SSR allele calling, and model parameters for STRUCTURE—are missing. Without these details, the work is not fully reproducible. Additionally, the manuscript inconsistently refers to PCA and PCoA, with one labeled in the figure and the other described in the text. Given the nature of the data (categorical, multi-allelic), PCoA is more appropriate, and this should be clearly stated and supported with references.

Validity of the findings

The conclusions currently overstate what the data support, particularly regarding population differentiation between “breeding lines” and “registered varieties.” These classifications appear to be administrative rather than biologically meaningful and may not reflect actual genetic divergence. The manuscript does not provide a rationale for why these groupings would be expected to exhibit population structure, especially given the use of highly polymorphic, neutral SSR markers.

The reported Fst value (0.098) indicates moderate, not strong, differentiation, yet the conclusions suggest a much higher degree of divergence than the data justify. Furthermore, the STRUCTURE results indicate K=2 is the most supported number of clusters, yet the manuscript focuses interpretation on K=4 without clear justification.

If the goal is to characterize genetic structure and diversity in cannabis, the authors are encouraged to explore grouping samples by biologically meaningful traits such as chemotype (e.g., Type I–IV), plant use (hemp vs. drug-type), or breeding system (dioecious vs. monoecious). These categories are more likely to reflect true evolutionary and breeding histories.

Additionally, the foundational microsatellite data need to be revisited. Several loci show allele sizes inconsistent with their repeat unit length, suggesting binning errors that may invalidate downstream analyses. Until this is corrected and raw allele data are transparently provided, the findings cannot be considered robust or fully supported by the underlying data.

Additional comments

Document with details attached.

Annotated reviews are not available for download in order to protect the identity of reviewers who chose to remain anonymous.

·

Basic reporting

I thank the authors for improving the manuscrupt following the suggested comments.

The manuscript is now comprehensive of the prevously lacking part, mainly related to the methodologies adopted and to the analyses procedures made. the research is now clear and, in my opinion, needs no further revisions from the authors in the content of the manuscript.

I only suggest authors to check figures to verify they are all correct in the formating of the text.

Experimental design

'no comment'

Validity of the findings

'no comment'

---

## Round 0.3 · Major Revisions

· Academic Editor

Major Revisions

Dear colleague, the reviewer has assessed your manuscript and requires another round of revisions in order to address several caveats regarding chemotypes and statements throughout the manuscript. please also consider the annotated manuscript in your rebuttal

Reviewer 1 ·

Basic reporting

The manuscript has improved in presentation, particularly in the updated figures, which are clearer and more appropriately labeled. However, there are several areas where basic reporting still needs attention. The writing requires tightening for clarity and precision, particularly in the Introduction and Discussion. Some terms (e.g., "genomic differences") are used inaccurately, and statements are occasionally overstated or speculative without sufficient support.

The background generally covers relevant prior work, but the classification of samples by chemotype (Types I–IV) is presented as fact without cannabinoid data, which is problematic given the inclusion of grain and fiber cultivars that may not produce measurable levels of cannabinoids. Literature citations are used inconsistently — some references support key points well, while others are used inappropriately to justify speculative claims.

The structure of the article follows standard format, and figures and tables are appropriate, though some interpretations (e.g., of PCoA and STRUCTURE plots) are overstated relative to the visual and statistical support. The manuscript would benefit from a clearer statement of limitations and greater transparency about how certain groupings (e.g., chemotype, population) were assigned.

Experimental design

The research falls within the scope of the journal and presents a relevant and meaningful question related to genetic diversity and sex determination in Cannabis sativa. However, the study does not clearly define its knowledge gap, and some of its framing (e.g., on chemotype categorization and population structure) implies broader genomic conclusions than are supported by the use of 10 SSR markers.

The methods are generally appropriate, but require more clarity in several areas, including how chemotype assignments were made, how STRUCTURE results were interpreted, and how sex identification results were validated. The authors reference 100% accuracy in sex determination via HRM, which is an overstatement and should be tempered.

While the SSR approach is valid, the scope and resolution are limited, and should be framed as such. The lack of chemical or phenotypic data weakens the conclusions drawn about population types and chemotypes.

Validity of the findings

The findings appear statistically valid, but are at times overstated. Nei’s Genetic Distance and Identity are misinterpreted as within-group metrics, when they are properly applied as pairwise measures between populations. Claims about genome-wide divergence, admixture, and domestication history are not well supported by the resolution of the data (10 SSRs), and should be revised or removed.

The conclusions regarding population structure, sex determination, and diversity are broadly supported, but require more precise and cautious language. The manuscript also lacks a clear discussion of limitations, which is especially important given the small sample size, limited marker set, and lack of phenotypic data.

Additional comments

This manuscript addresses a relevant topic, and improvements to the figures are appreciated. However, several important issues remain regarding interpretation, clarity, and classification strategies. I recommend major revisions before the manuscript is suitable for publication.

Details are attached in a Word document for the authors to consult. This includes specific comments and suggested revisions throughout the manuscript to guide improvements in scientific accuracy, structure, and presentation.

Annotated reviews are not available for download in order to protect the identity of reviewers who chose to remain anonymous.

---

## Round 0.4 · Minor Revisions

· Academic Editor

Minor Revisions

Dear colleagues, the reviewer has completed the assessment of your manuscript and suggested it has merit for publication following minor revisions. Please find the comments in the attached annotated file.

Reviewer 1 ·

Basic reporting

The manuscript is generally well written in professional English, with appropriate structure, figures, and references. It provides sufficient background to situate the study within the broader field of cannabis genetics. While much improved from earlier drafts, some minor issues remain, including formatting inconsistencies, unclear phrasing, and insufficient interpretation in parts of the discussion. I have provided detailed, line-by-line suggestions to address these points.

Experimental design

The study presents original research that fits well within the journal’s scope and addresses a relevant and timely question in cannabis genetics. The research question is clearly stated, and the manuscript identifies a meaningful gap in our understanding of genetic differentiation between hemp and marijuana.
The investigation appears to be conducted rigorously and meets technical and ethical standards for the field, although there was a question about the cannabinoids tested which you will find in my detailed review. Methods are described in sufficient detail to allow for reproducibility, with appropriate references and descriptions of sample processing, genetic analysis, and data interpretation.

Validity of the findings

The study presents a meaningful contribution to the literature by providing new data on genetic variation between hemp and marijuana, a topic of ongoing interest and relevance. While not aiming for high conceptual novelty, the work fills an identified gap with a clearly stated rationale and sound methodology. The underlying data appear robust, statistically appropriate, and well controlled. Conclusions are clearly linked to the research question and remain within the scope of the results without overstating causality.

Additional comments

I have provided a line by line review with minor changes.

Annotated reviews are not available for download in order to protect the identity of reviewers who chose to remain anonymous.

---

## Round 0.5 · Minor Revisions

· Academic Editor

Minor Revisions

Dear colleagues, despite the significant improvements of your manuscript, the reviewer requires a minor revision. Please address the raised points in the annotated file

Reviewer 1 ·

Basic reporting

Minor formatting and spacing issues remain throughout the manuscript (e.g., inconsistent italics, paragraph breaks, spacing between words). Detailed notes are included in the attached document.
Additional clarification is needed around the distinction between chemovar/type classification and neutral marker-based population structure. Figure interpretations, particularly STRUCTURE, PCoA, and UPGMA, are not fully aligned with what the data show. STRUCTURE plots are ordered alphabetically, obscuring genetic clustering. The study interpretation is constrained by a chemovar framework that is not supported by the SSR data. The STRUCTURE results at K=4 reveal population structure that is not meaningfully explored. This disconnect should be addressed for the results to fully support the stated hypotheses.

Experimental design

The study applies appropriate methods, and the experimental setup is generally sound. However, the interpretation relies too heavily on chemovar groupings (Types I–V), which are based on functional traits not captured by neutral SSR markers. This limits the strength of the conclusions. The exclusion of CBGA and CBCA from cannabinoid profiling is also odd. These are easily detected by LC-MS, and the rationale provided doesn’t align with analytical norms or regulatory relevance.

Validity of the findings

The data and analyses are sound, but the interpretation oversimplifies key results. STRUCTURE clearly supports four clusters, yet conclusions default to a binary grouping.The mention of a multidimensional framework is promising but not reflected in the actual analysis.

Additional comments

After multiple rounds of revision, the manuscript is closer to being publication-ready, but several key issues remain. The continued reliance on a predefined “Type I–V” chemovar framework still limits the interpretation of results and does not fully reflect what the SSR data reveal. The STRUCTURE results clearly support four distinct genetic clusters, yet this is largely reduced to a binary comparison in the narrative. There are also persistent formatting inconsistencies throughout. A final round of careful revision, both in framing and presentation, would significantly improve the manuscript.

Annotated reviews are not available for download in order to protect the identity of reviewers who chose to remain anonymous.

---

## Round 0.6 · accepted · Accept

· Academic Editor

Accept

Dear colleagues, after examining your revised work, all minor comments have been thoroughly addressed, and therefore, your manuscript should be accepted as it stands after four revision rounds.